# Gain of Aggressive Histological and Molecular Patterns after Acquired Resistance to Novel Anti-EGFR Therapies in Non-Small Cell Lung Cancer

**DOI:** 10.3390/ijms24043802

**Published:** 2023-02-14

**Authors:** Clémence Basse, Olfa Trabelsi-Grati, Julien Masliah, Céline Callens, Maud Kamal, Paul Freneaux, Jerzy Klijanienko, Ivan Bieche, Nicolas Girard

**Affiliations:** 1Institut du Thorax Curie Montsouris, 75005 Paris, France; 2Paris Saclay Campus, Versailles Saint Quentin University, 78000 Versailles, France; 3Genetics Department, Institut Curie, 75005 Paris, France; 4Department of Drug Development and Innovation, Institut Curie, 75005 Paris, France; 5Pathology Department, Institut Curie, 75005 Paris, France; 6Genetics Department, University Paris Descartes, 75005 Paris, France

**Keywords:** novel anti-EGFR, resistance, lung adenocarcinoma, SCLC-like, RB1, TP53

## Abstract

Novel anti-EGFR therapies target resistance to standard-of-care anti-EGFR in patients with metastatic lung cancer. We describe tumors at progression versus at the initiation of novel anti-EGFR agents in patients with metastatic lung adenocarcinoma harboring EGFR mutation. This clinical case series reports the histological and genomic features and their evolution following disease progression under amivantamab or patritumab-deruxtecan in clinical trials. All patients had a biopsy at disease progression. Four patients harboring EGFR gene mutations were included. Three of them received anterior anti-EGFR treatment. Median delay to disease progression was 15 months (range: 4–24). At progression, all tumors presented a mutation in the TP53 signaling pathway associated with a loss of heterozygosis (LOH) of the allele in 75% (*n* = 3), and two tumors (50%) presented an RB1 mutation associated with LOH. Ki67 expression increased above 50% (range 50–90%) in all samples compared to baseline (range 10–30%), and one tumor expressed a positive neuroendocrine marker at progression. Our work reports the potential molecular mechanisms of resistance under novel anti-EGFR in patients with metastatic EGFR-mutated lung adenocarcinoma, with the transformation to a more aggressive histology with acquired TP53 mutation and/or the increase in Ki67 expression. These characteristics are usually found in aggressive Small Cell Lung Cancer.

## 1. Introduction

### 1.1. Context

Non-small cell Lung cancer (NSCLC) accounts for 85–90% of all lung cancers, adenocarcinoma representing the main histology [1]. A mutation in the *EGFR* (Epidermal Growth Factor Receptor) gene is found in 15% of adenocarcinoma [2]. Common *EGFR* mutations are exon 19 deletion and exon 21 L858R point mutation. They account for 85% of *EGFR* mutations, and they confer sensitivity to EGFR tyrosine kinase inhibitors (TKIs) [3]. Although the majority of these patients initially respond to TKIs, they then acquire resistance, preventing durable response. Today third-generation TKIs represent the standard-of-care in the first-line setting of metastatic lung adenocarcinoma (LUAD) harboring common *EGFR* mutation. However, complex and diverse molecular mechanisms of resistance have been observed, including *MET* (*MET* proto-oncogene) dysregulation in 15% of cases, and additional acquired *EGFR* mutations [4,5]. The uncommon *EGFR* exon 20 insertion mutation (Exon20ins) is less frequent and found in 9–12% of *EGFR*-mutated NSCLC. This latter alteration confers primary resistance to EGFR TKIs. Patients with NSCLC harboring an Exon20ins are not offered EGFR TKIs in routine care [6], but novel anti-EGFR therapies are under development in this setting.

Small cell lung cancers (SCLC) represent 15% of lung cancer and have a worst prognosis than NSCLC [1]. They are known to harbor *RB1* mutations in almost all cases and to have an alteration in the TP53 signaling pathway [7].

Amivantamab is a fully human immunoglobulin G1 (IgG1)-based bispecific antibody directed against the EGF and cMet receptors, administered in the phase 1 CHRYSALIS trial (NCT02609776) [8]. Amivantamab was administered in this trial at progression after platin-doublet-based chemotherapy in patients with NSCLC harboring an *EGFR* exon 20 insertion mutation, leading to primary resistance to all anti-EGFR TKIs. Patritumab-deruxtecan is an antibody drug conjugate comprising a recombinant fully human anti-human epidermal growth factor receptor 3 (HER3) IgG1 monoclonal antibody covalently linked to a tetrapeptide linker containing a topoisomerase I inhibitor, administered in the phase 2 HERTEHNA trial (NCT04619004) [9]. Patritumab-deruxtecan was administered at progression after one or two EGFR TKIs.

Even when these innovative anti-EGFR therapies have promising results, some patients still present disease progression.

### 1.2. Objective

Our work is a retrospective case series of patients with metastatic lung cancer treated in clinical trials with either amivantamab or patritumab-deruxtecan who presented disease progression under novel ant-EGFR agents. Our objective is to describe the pathological and molecular features of tumor specimens at progression versus at treatment initiation, to identify the potential mechanisms of resistance to novel anti-EGFR therapies. This could help in the choice of subsequent treatments.

### 1.3. Method

We included patients with *EGFR*-mutated NSCLC, enrolled, either in the phase 1 CHRYSALIS trial or in the phase 2 HERTHENA trial, implemented from 1 January 2020 at our institute. Patients were consecutively included in our case series whenever they were experiencing disease progression. The cutoff date was 31 December 2021. All patients gave their informed consent for this report. 

A biopsy was performed at baseline per routine care, and a supplementary biopsy was systematically performed at disease progression. All samples were directed to pathological review.

Paraffin sections of lung biopsies were analyzed by two independent pathologists, who performed a blinded review of the samples. Immunohistochemistry for neuroendocrine differentiation (CD56, synaptophysin, chromogranin A) and the Ki67 marker of proliferation were scored.

Concerning the NGS method, samples were systematically assessed for quality control, which included verification of sufficient cell content >30% in the biopsies (quality criteria used in routine care in clinical trials). Samples were not additionally processed before sequencing so as to not alter the analysis. Sequencing was carried out using an in-house large capture-based targeted next-generation sequencing panel of 571 genes, called DRAGON Dx (Detection of Relevant Alterations in Genes involved in Oncogenetics). This NGS panel has been developed by the genetics department of Institut Curie (Paris, France) and can detect mutations, copy number alterations (CNA), tumor mutational burden, and microsatellite instability. It is composed of 571 genes of interest in oncology for diagnosis, prognosis, and theragnostics. The whole method is described in different papers and allow detection of a larger panel of mutations than can be detected in routinely-used NGS panels [10]. NGS libraries are prepared using an Agilent SureSelect XT-HS kit, Agilent Technologies France, Les Ulis, France. Following this, sequencing was performed on NovaSeq (Illumina) at the Institut Curie core genomic facility, with a mean reading depth of 1500× and a minimal depth of 300×. Variant calling was performed using Varscan2 (v2.4.3) (RRID:SCR006849)

## 2. Detailed Case Description

### 2.1. Patient Information and Molecular Profile of the Initial Tumor

Four patients with metastatic lung adenocarcinoma were included. They were harboring *EGFR* gene mutations; exon 20 insertion (*n* = 2), *EGFR* exon 21 L858R mutation (*n* = 1), co-occurring *EGFR* exon 19 deletion, and exon 20 T790M mutation (*n* = 1) were included. All were women (100%). At the time of inclusion in the clinical trial, Patient#1 (Tumor#1, harboring exon21 *EGFR* L858R mutation) had received two previous lines of treatment with one third-generation TKI (osimertinib and platin-doublet chemotherapy); Patient#2 (Tumor#2, harboring *EGFR* exon 20 insertion) had received one line of platin-doublet chemotherapy); Patient#3 (Tumor#3, harboring del19 and T790M *EGFR* mutations) had received four anterior lines of treatment with two TKIs: erlotinib (first-generation), then osimertinib (third-generation) for the T790M acquired resistance mutation under erlotinib, and two lines of platin-doublet chemotherapy; Patient#4 (*EGFR* exon 20 insertion) had received two lines of platin-doublet chemotherapy and 1 TKI afatinib (second-generation) before starting the experimental treatment. The median delay to progression disease was 15 months (range: 4–24 months), Table 1. Patient#3 had the shorter progression-free survival under experimental treatment (4 months) and was the higher pretreated patient from the cohort (the only patient with two previous TKIs). He was also the only one with the transformation in undifferentiated large cell aggressive neuroendocrine carcinoma in histology. Sites of progression that were biopsied were in all cases pre-existing known tumoral sites (lung, bone, and liver).

### 2.2. Pathological Findings: Differentiation in a More Aggressive Histology

All four tumors were morphologically well differentiated lung adenocarcinoma at baseline. At progression, one tumor transformed in an undifferentiated large cell neuroendocrine carcinoma (Tumor#3), and all samples increased the Ki67 expression above 50% (range 50–90%), compared with the expression at baseline (range 10–30%); Table 1, Figure 1.

### 2.3. Molecular Profile of the Tumor at Progression: Alteration in TP53 and RB1 Signaling Pathway

All initial *EGFR* mutations were still found at progression with a similar allele frequency (AF), except in one patient: Tumor#2 presented a gain on the mutated *EGFR* allele (allele frequency of the EGFR exon 20 insertion reaching 51% at progression versus 10% at baseline), Table 1.

An alteration in the TP53 signaling pathway was found in all tumors at progression. In three tumors, this was conducted to an inactivation of the *TP53* gene: mutation c.920-1G>A/p.? with an AF of 52% associated with an LOH, (Tumor#1); mutation c.301A>T/p.(Lys101*) with an AF of 37% (Tumor#2); and mutation c.395A>G/p.(Lys132Arg) with an AF of 75% associated with an LOH (Tumor#3). The *TP53* mutation was already present at baseline in Tumor#3. In Tumort#4, a c.2702-4 del, p.? TP53BP1 mutation with an AF of 12% was found.

Two patients (Tumor#1, Tumor#2) acquired a mutation leading to an inactivation of the *TP53* gene at progression, and also acquired an *RB1* gene alteration at progression: c.2106+1G>A/p? mutation (AF = 41%), also associated with an LOH, and c.1389+1_1389+2del/p.? mutation (AF = 35%) associated with an LOH, respectively; Figure 2.

The patient with a *TP53* mutation at baseline and at progression (Tumor#3) was not harboring an alteration of the *RB1* gene at progression. However, Tumor#3 was an undifferentiated large cell carcinoma with a positive expression of neuroendocrine markers at progression, suggesting a transformation to a more aggressive histology, and a supplementary step towards a differentiation in SCLC-like tumors; Table 1, Figure 1.

### 2.4. Transformation under Novel Anti-EGFR Therapies Patient by Patient

Patient#1. The histological transformation was mainly the elevation of Ki67 expression from 30% at baseline towards 60% at progression. The phenotype description was comparable to a differentiated adenocarcinoma at baseline and at progression. Neuroendocrine markers were not seen at progression. Concerning molecular alterations, Tumor#1 presented an *RB1* LOH and a *TP53* LOH at progression, these two alterations not being found at baseline. This suggests either a pressure on the more proliferative tumoral clone under patritumab-deruxtecan, and/or the development of acquired mutations.

Patient#2. The histological transformation consisted of the elevation of Ki67 expression (from 20% at baseline towards >90% at progression). The histological phenotype remained comparable at baseline and at progression to a differentiated adenocarcinoma. Neuroendocrine markers were not seen at progression. Concerning molecular alteration, Tumor#2 presented an *RB1* mutation with LOH as well as a *TP53* mutation with LOH at progression; these two alterations were not found at baseline.

Patient#3. This is the only patient who presented with true histological subtype transformation, from a differentiated adenocarcinoma at baseline towards an undifferentiated large cell carcinoma with positive neuroendocrine markers at progression (positivity of chromogranin A and synaptophysin markers at progression). We also noticed an elevation of the Ki67 expression from 10% at baseline towards 90% at progression. Concerning molecular alteration, Tumor#3 did not harbor a *RB1* alteration, but presented a *TP53* mutation with LOH at baseline and at progression. This result suggests that the transformation in more aggressive neuroendocrine tumors could be explained by the pre-existence of an aggressive clone that was positively selected by the novel therapy (here, patritumab-deruxtecan).

Pattent#4. The histological transformation noted in the patients was an elevation of the Ki67 proliferative marker expression (from 10% at baseline versus 50% at progression). Neuroendocrine markers were not seen at progression. Concerning molecular alteration, Tumor#4 harbored a mutation in the *TP53BP1* gene.

## 3. Discussion

Our report is the first to provide insights into the resistance mechanisms to novel anti-EGFR agents in NSCLC. Here we identified the emergence of more aggressive and proliferative tumor features at progression: an increased expression of the Ki67 marker, the presence of inactivating mutations in the *RB1* tumor suppressor gene, and alteration in the TP53 signaling pathway. These results suggest the acquisition of multiple alterations in our patients towards the SCLC-like tumor phenotype. As previously reported, the pathway towards the SCLC phenotype would start with the acquisition of *TP53* and *RB1* loss of function [11,12], followed by an increased expression of Ki67, followed by the acquisition of the neuro-endocrine markers in favor of the histological transformation in neuro-endocrine tumors [13]; Figure 3. Acquired *RB1* suppression has previously been reported after progression under EGFR tyrosine kinase inhibitors in patients with common *EGFR* mutations [14]. Concerning Patient#4, with a tumor harboring a *TP53* mutation at baseline, progression-free survival was the shortest under the experimental treatment. It has been reported that concomitant the *TP53* mutation confers worse prognosis in EGFR-mutated NSCLC patients treated with TKIs [15].

In line with our findings, Offin et al. recently published that patients affected by triple EGFR/TP53/RB1-mutant lung cancers are at risk of histologic transformation, with 25% presenting with de novo SCLC or SCLC transformation [16]. SCLC transformation represents a known mechanism of resistance to osimertinib (third-generation EGFR TKIs) in EGFR-mutated lung adenocarcinoma, which dramatically impacts patients’ prognosis due to high refractoriness to conventional treatments [17].

## 4. Conclusions

Our case series is the first, to our knowledge, to report the molecular and pathological mechanisms of resistance to amivantamab and patritumab deruxtecan, novel and innovative anti-EGFR therapies. Histological transdifferentiation to a more aggressive histology and towards SCLC-like transformation was identified in one of our patients that subsequently render the patient eligible for chemotherapy, based on the platin and etoposide indicated in this histology. The profound inhibition of the EGFR, MET, and HER3 signaling pathways associated with amivantamab and patritumab deruxtecan makes it likely for such complex molecular resistance mechanisms to emerge. We highly recommend rebiopsy being performed, if possible, in patients developing acquired resistance to novel anti-EGFR targeted therapies to help understand the underlying mechanisms of resistance. This could help in the choice of subsequent treatment.

## Figures and Tables

**Figure 1 ijms-24-03802-f001:**
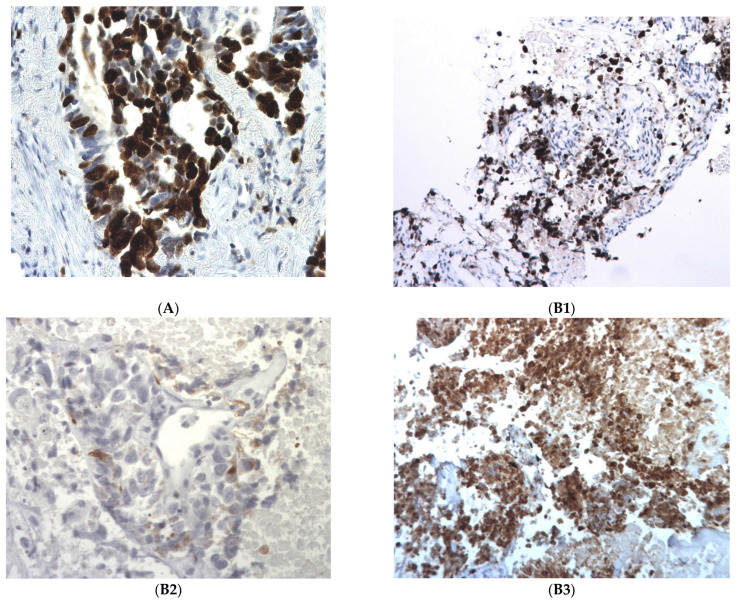
Immunohistochemistry patterns in post-therapeutic biopsies. (**A**) Tumor#2. High Ki67, estimated to be more than 90%. Tumor#3. (**B1**) Tumor#3. High Ki67, estimated to be more than 90%; (**B2**) Tumor#3. Slight Chromogranin A positivity; (**B3**) Tumor#3. Strong synaptophysin positivity.

**Figure 2 ijms-24-03802-f002:**
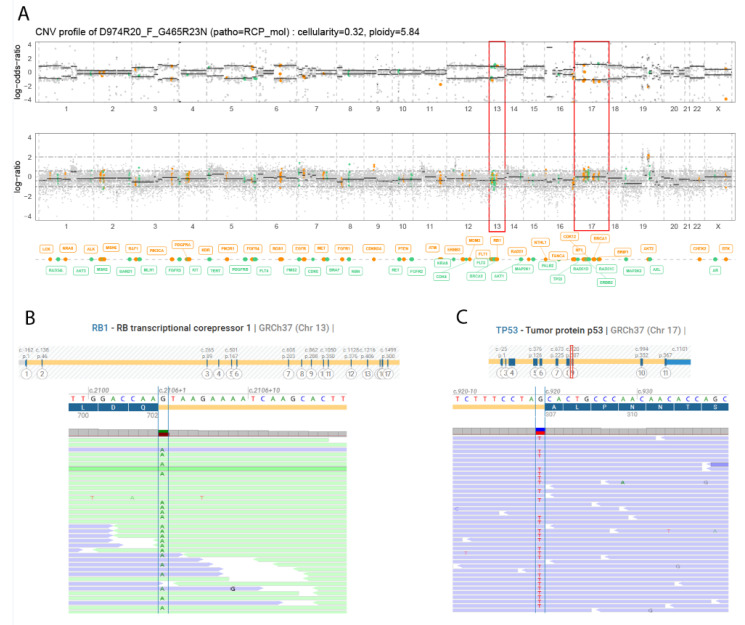
Acquired inactivation of *TP53* and *RB1* genes at progression in Tumor#1. (**A**) Genomic profiling showing loss of heterozygosis of *TP53* (isodisomy of chromosome 17) and *RB1* genes (isodisomy of chromosome 13). (**B**) Splicing *RB1* c.2106+1G>A mutation. (**C**) Splicing *TP53* c.920-1G>A mutation.

**Figure 3 ijms-24-03802-f003:**
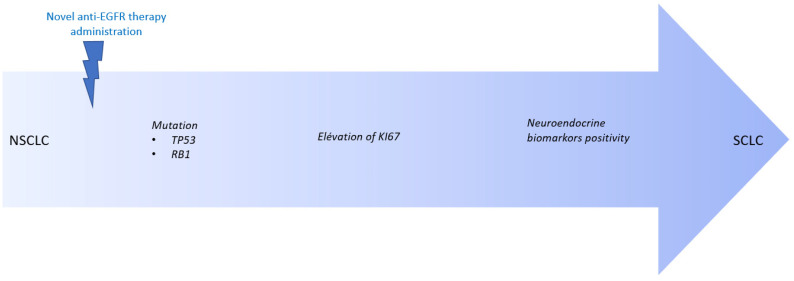
Transformation from NSCLC to SCLC under novel anti-EGFR treatment. Legend: NSCLC = Non-Small Cell Lung Cancer; SCLC = Small Cell Lung Cancer.

**Table 1 ijms-24-03802-t001:** Comparison of molecular and histological features at baseline versus at progression under novel anti-EGFR therapies. LUAD = Lung adenocarcinoma; AF = allelic frequency; LOH = Loss of heterozygocy.

#	Age	Molecular Characteristics at Baseline	Pathological Characteristics at Baseline before Treatment	Treatment Administered	Delay under Treatment before Progression	Molecular Characteristics at Progression	Pathological Characteristics at Progression after Treatment
1	56	Exon 21 EGFR L858R (c.2573T>G), AF of 52%	Differenciated LUAD Ki67 30% Chromogranin A − CD56 − Synaptophysin −	patritumab deruxtecan	15 months	EGFR c.2573T>G/p.(Leu858Arg), AF of 35% RB1 c.2106+1G>A/p.?, AF of 41% with LOH TP53 c.920−1G>A/p.?, AF of 52% with LOH	Differenciated LUAD Ki67 60% Chromogranin A − CD56 − Synaptophysin −
2	66	EGFR exon 20 Insertion, c.2313_2314insGTC/p.(Asn771_Pro772insVal), AF of 10%	Differenciated LUAD Ki67 20% Chromogranin A − CD56 − Synaptophysin −	amivantamab	17 months	EGFR exon 20 Insertion, c.2313_2314insGTC/p.(Asn771_Pro772insVal), AF of 51% RB1 mutation c.1389+1_1389+2del/p.? (AF of 35%), heterozygocy with LOH TP53 mutation c.301A>T/p. (Lys101*), AF of 37% with LOH RBM10 mutation c.2398G>T/p.(Glu800*), AF of 24%	Differenciated LUAD Ki67 > 90% Chromogranin A − CD56 − Synaptophysin −
3	58	Del 19 EGFR c.2240_2257del/p., AF of 49% EGFR c.2369C>T/p.(Thr790Met), AF of 12% TP53 mutation c.395A>G (p.Lys132Arg) (AF of 90%) with LOH	Differenciated LUAD Ki67 10% Chromogranin A − CD56 − Synaptophysin −	patritumab deruxtecan	4 months	Del 19 EGFR c.2240_2257del/p., AF of 34.2% EGFR c.2369C>T/p.(Thr790Met), AF of 9% TP53 c.395A>G/p.(Lys132Arg), AF of 75% with LOH MYC amplification PIK3CB c.3151G>A/p.(Glu1051Lys), AF of 59%)	Undifferenciated Large cell carcinoma Ki67 90% Chromogranin A + CD56 – Synaptophysin +
4	66	EGFR exon 20 Insertion c.2284-5_2290dup/p.(Ala763_Tyr764insPheGlnGluAla), AF of 18%	Differenciated LUAD Ki67 10% Chromogranin A − CD56 − Synaptophysin −	amivantamab	24 months	EGFR exon 20 Insertion c.2284-5_2290dup/p. p.(Ala763_Tyr764insPheGlnGluAla), AF of 16% FGFR1 c.1731C>A/p.(Asn577Lys), AF of 2% TP53BP1 mutation c.2702-4del, p.?, AF of 12%	Differenciated LUAD Ki67 50% Chromogranin A − CD56 − Synaptophysin −

## Data Availability

No new data were created. Data are available in patients medical records.

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
