# Peer review of "Gain of Aggressive Histological and Molecular Patterns after Acquired Resistance to Novel Anti-EGFR Therapies in Non-Small Cell Lung Cancer"

_ijms, 2023, doi:10.3390/ijms24043802_

Round 1

Reviewer 1 Report

In this paper authors described potential resistance mechanisms to novel anti-EGFR agents in lung adenocarcinomas harbouring EGFR mutations.

The topic is interesting, but some aspects have to be clarified:

- The expression “Risk of SCLC transformation under novel anti EGFR therapies” should be removed from the title, considering that only four patients are described and only one, besides TP53 and RB1 mutations, finally showed the histological transformation from adenocarcinoma to undifferentiated large cell carcinoma.

- Authors should clearly specify the line of treatment for the four patients enrolled in the two trials CHRYSALIS and HERTEHNA and report also previous treatments (TKI rather than chemotherapy, etc.) and related progression time and resistance mechanisms.

-One case harboured both EGFR exon 19 deletion and the resistance mutation T790M, is the T790M treatment naïve or has it been aquired after a previous line TKI treatment? Please specify and discuss.

- One case had TP53 mutation at baseline, authors should discuss, according to literature data, how the co-occurrence of TP53 and EGFR activating mutations can impact on response to therapy.

Author Response

In this paper authors described potential resistance mechanisms to novel anti-EGFR agents in lung adenocarcinomas harbouring EGFR mutations.

The topic is interesting, but some aspects have to be clarified:

- The expression “Risk of SCLC transformation under novel anti EGFR therapies” should be removed from the title, considering that only four patients are described and only one, besides TP53 and RB1 mutations, finally showed the histological transformation from adenocarcinoma to undifferentiated large cell carcinoma.

Thank you for this comment. We modified the Title “Risk of transformation in more aggressive histology under novel anti EGFR therapies”

- Authors should clearly specify the line of treatment for the four patients enrolled in the two trials CHRYSALIS and HERTEHNA and report also previous treatments (TKI rather than chemotherapy, etc.) and related progression time and resistance mechanisms.

Thank you for this relevant point. We modified the first paragraph concerning patients information in the Results part as follow:

Four patients with metastatic lung adenocarcinoma were included. They were harboring EGFR gene mutations: exon 20 insertion (n=2), EGFR exon 21 L858R mutation (n=1), and co-occuring EGFR exon 19 deletion and exon 20 T790M mutation (n=1) were included. All were women (100%). At the time of inclusion in the clinical trial, Patient#1 (Tumor#1 harboring exon21 EGFR L858R mutation) had received 2 previous lines of treatment with 1 TKI (osimertinib and platin-doublet chemotherapy); Patient#2 (Tumor#2 harboring EGFR exon20insertion) had received one line of platin-doublet chemotherapy); Patient#3 (Tumor#3 harboring del19 and T790M EGFR mutations) had received 4 anterior lines of treatment with 2 TKIs: erlotinib, then osimertinib for T790M acquired resistance mutation under erlotinib, and 2 lines of platin-doublet chemotherapy; Patient#4 (EGFR exon20 insertion) had received 2 lines of platin-doublet chemotherapy and 1 TKI afatinib before starting the experimental treatment. The median delay of progression disease was 15 months [range: 4-24 months], Table 1. Patient#3 had the shorter progression free survival under experimental treatment (4 months), and was the higher pretreated patient from the cohort (the only patient with previous 2 TKIs). He was also the only one with the transformation in undifferentiated large cell aggressive neuroendocrine carcinoma in histology. Sites of progression that were biopsied were in all cases pre-existing known tumoral sites (lung, bone and liver).

-One case harboured both EGFR exon 19 deletion and the resistance mutation T790M, is the T790M treatment naïve or has it been acquired after a previous line TKI treatment? Please specify and discuss.

Thank you for your question, we answered in the previous question.

- One case had TP53 mutation at baseline, authors should discuss, according to literature data, how the co-occurrence of TP53 and EGFR activating mutations can impact on response to therapy.

Thank you for this comment. We added in the discussion the following text: Concerning Patient#4 with a tumor harboring a TP53 mutation at baseline, progression free survival was the shortest of the case series under the experimental treatment. It has been reported that concomitant TP53 mutation confers worse prognosis in EGFR-mutated NSCLC patients treated with TKIs [15].

Reviewer 2 Report

The case study presented by the Clémence BASSE et al. provides the molecular and pathological mechanisms of resistance to amivantamab and patritumab deruxtecan in EGFR-positive metastatic NSCLC. However, the sequencing result was not confirmed by any other gold standard method. Thus it remains a possibility that some of the acquired alterations were gained as artifacts. Especially, the authors did not provide information if the tested samples were additionally processed by pathologists before sequencing, which may affect the results. In the end, the main conclusions were provided on analysis of only 4 cases limited to the female gender, and there was no negative control analyzed. Based on these limitations it is strongly suggested to be more careful with the final conclusions.

Minor remarks

1. The title is very confusing to the reader.

2. The paper is full of typos and spelling errors. I recommend proofreading by a native speaker.

3. I would suggest remodelling the introduction part. Its two first paragraphs are limited to well know data, while the third one, which seems to be essential to the study, is very generic. For instance, adding the indications for the administration of the new drugs would be very informative.

4. Quality of the figure 2 is very poor and its resolution needs to be improved.

5. All the figures and tables should be proofread.

6. The idea of showing tumors of two different patients in figure 1 is unclear. The case study is limited only to 4 cases, it would be grateful to describe all of them one by one and then provide the final conclusion.

7. For all abbreviations in the text, tables, and figures please follow TCGA nomenclature

8. Please add the CHRYSALIS trial number.

Author Response

The case study presented by the Clémence BASSE et al. provides the molecular and pathological mechanisms of resistance to amivantamab and patritumab deruxtecan in EGFR-positive metastatic NSCLC. However, the sequencing result was not confirmed by any other gold standard method. Thus it remains a possibility that some of the acquired alterations were gained as artifacts. Especially, the authors did not provide information if the tested samples were additionally processed by pathologists before sequencing, which may affect the results. In the end, the main conclusions were provided on analysis of only 4 cases limited to the female gender, and there was no negative control analyzed. Based on these limitations it is strongly suggested to be more careful with the final conclusions.

We thank you for these comments and precautions.

Concerning the NGS method, samples were systematically assessed for quality control, that included verification of sufficient cell content >30% in the biopsies. Samples were not additionally processed by pathologists before sequencing. The NGS panel (called “DRAGON”) has been developed by the genetics department of Institut Curie (Paris, France). It can detect mutations, copy number alterations (CNA), tumor mutational burden and microsatellite instability. It is composed of 571 genes of interest in oncology for diagnosis, prognosis, and theragnostics. The whole method is described in different paper [10].

We adapted the conclusion according to this remark “Histological transdifferentiation to more aggressive histology and towards SCLC-like transformation was identified in one of our patients that subsequently render the patient eligible to chemotherapy based on platin and etoposide »

Minor remarks

  1. The title is very confusing to the reader.

We modified the title in « Small cell lung cancer-like histological and molecular features associated with acquired resistance to novel anti-EGFR therapies in metastatic EGFR mutated lung cancer: Risk of transformation in more aggressive histology under novel anti EGFR therapies”

  1. The paper is full of typos and spelling errors. I recommend proofreading by a native speaker.

Thank you for your comment, we revised the manuscript.

  1. I would suggest remodelling the introduction part. Its two first paragraphs are limited to well know data, while the third one, which seems to be essential to the study, is very generic. For instance, adding the indications for the administration of the new drugs would be very informative.

This is a relevant remark, here is the modified text: Amivantamab is a fully human immunoglobulin G1 (IgG1)-based bispecific antibody directed against the EGF and cMet receptors administered in the phase 1 CHRYSALIS trial (NCT02609776) [8]. Amivantamab was administered in this trial at progression after platin-doublet based chemotherapy in patients with NSCLC harboring an EGFR exon 20 insertion mutation, leading to primary resistance to all anti-EGFR TKIs. Patritumab-deruxtecan is an antibody drug conjugate comprising a recombinant fully human anti-human epidermal growth factor receptor 3 (HER3) IgG1 monoclonal antibody covalently linked to a tetrapeptide linker containing a topoisomerase I inhibitor, administered in the phase 2 HERTEHNA trial (NCT04619004) [9]. Patritumab-deruxtecan was administered at progression after one or two EGFR TKIs.

If these innovative anti EGFR therapies have promising results, some patients still present disease progression.

  1. Quality of the figure 2 is very poor and its resolution needs to be improved.

We modified it.

  1. All the figures and tables should be proofread.

We reviewed all the figures and tables.

  1. The idea of showing tumors of two different patients in figure 1 is unclear. The case study is limited only to 4 cases, it would be grateful to describe all of them one by one and then provide the final conclusion.

Thank you for your comment. We choose to show in Figure 1 telling examples of the transdifferenciation in aggressive histological subtype. Concerning the description of the 4 cases we added a paragraph with a description of each patient as suggested:

Transformation under novel anti-EGFR therapies patient by patient

Patient#1. The histological transformation was mainly the elevation of Ki67 expression from 30% at baseline towards 60% at progression. The phenotype description was compatible with a differentiated adenocarcinoma at baseline and at progression. Neuroendocrine markers were not seen at progression. Concerning molecular alterations, Tumor#1 presented with RB1 LOH and TP53 LOH at progression, these two alterations being not found at baseline. This suggests either a pressure on the more proliferative tumoral clone under patritumab-deruxtecan, and/or the development of acquired mutations.

Patient#2. The histological transformation was also the elevation of Ki67 expression from 20% at baseline towards >90% at progression. The histological phenotype remained compatible at baseline and at progression with a differentiated adenocarcinoma. Neuroendocrine markers were not seen at progression. Concerning molecular alteration, Tumor#2 presented with RB1 mutation with LOH as well as a TP53 mutation with LOH at progression, these two alterations being no found at baseline.

Patient#3. This is the only patient where a transformation in the histologcal subtype was observed, from a differentiated adenocarcinoma at baseline towards an undifferentiated large cell carcinoma with positive neuroendocrine markers (chromogranin A and synaptophysin) at progression. We also noticed an elevation of the Ki67 expression from 10% at baseline towards 90% at progression. Concerning molecular alteration, Tumor#3 did not harbor a RB1 alteration, but retained TP53 mutation with LOH. This result suggests that the transformation in neuroendocrine and aggressive tumor is not necessary related to the acquisition of molecular alterations specific to SCLC-like tumors. The aggressive tumoral clone was possibly already present at baseline and selected by patritumab-deruxtecan.

Pattent#4. The histological transformation was seen in the elevation of Ki67 expression from 10% at baseline versus 50% at progression. Neuroendocrine markers were not seen at progression. Concerning molecular alteration, Tumor#4 harbored a mutation in the TP53BP1 gene.

  1. For all abbreviations in the text, tables, and figures please follow TCGA nomenclature

We adapted our abbreviations according to the TCGA nomenclature.

  1. Please add the CHRYSALIS trial number.

This has been added.

Round 2

Reviewer 1 Report

Authors have improved their manusctript and  addressed the raised issues.

Author Response

We thank reviewer 1 for his positive comment. We have checked the English language again.

Reviewer 2 Report

The revised manuscript was slightly improved compared to the first version and did not follow the suggested methodological aspects. The paper cannot be accepted in its current form and should be submitted to a journal relevant to case reports.

Author Response

We thank Reviewer2 for his comment. We followed Reviewer2 comments in the second uploaded version of the text. We also improved the paper based on Reviewer1 recommendations.

To improve again the article, we revised the spelling, and the description of the patients one by one. We modified the title to not directly mention Small Cell Like Cancer. Our paper is original as we highlight some pathological and molecular features at cancer progression under novel targeted therapies in the setting of a rare subgroup of patients. This is an opportunity to show the importance and the role multimodal approach in this setting. It is very innovative, and represents the first reporting of such analyses after new anti-EGFR agents.

We hope the revision of the article will satisfy your request.